# Can Volunteering Buffer the Negative Impacts of Unemployment and Economic Inactivity on Mental Health? Longitudinal Evidence from the United Kingdom

**DOI:** 10.3390/ijerph19116809

**Published:** 2022-06-02

**Authors:** Senhu Wang, Wanying Ling, Zhuofei Lu, Yuewei Wei, Min Li, Ling Gao

**Affiliations:** 1Department of Sociology, National University of Singapore, 11 Arts Link, #03-06 AS1, Singapore 117573, Singapore; 2School of Sociology and Population Studies, Renmin University of China, No. 59 Zhongguancun Street, Beijing 100872, China; lingwany@ruc.edu.cn; 3Department of Social Statistics, University of Manchester, HBS Building, Oxford Road, Manchester M13 9PL, UK; 4Department of Sociology, University of Cambridge, 16 Mill Lane, Cambridge CB2 1SB, UK; yw547@cam.ac.uk; 5Independent Scholar, Guangli Road, Tianhe District, Guangzhou 510620, China; minli3134@163.com; 6School of Economics, Xiamen University, Siming South Road #422, Xiamen 361005, China

**Keywords:** volunteering, employment, gender, economically inactive groups, mental health

## Abstract

Previous research suggests that volunteering can mitigate the negative mental health impacts of unemployment but has yielded mixed results. This study extends the previous literature by examining whether volunteering can buffer the negative impacts of both unemployment and economic inactivity on mental health. Using nationally representative panel data from the United Kingdom Longitudinal Household Study (2010–2020) and fixed effects models, this study yields three important findings: First, volunteering cannot mitigate the adverse effects of unemployment, regardless of gender. Second, frequent volunteering (at least once per month) can benefit the mental health of economically inactive groups (e.g., family care and long-term sickness). Third, the study reveals the gendered patterns of the impacts of volunteering. Specifically, frequent volunteering can buffer the negative impacts of family care or long-term sickness for men, and the negative impacts of unpaid work for women. Overall, these findings contribute towards a more nuanced understanding of the buffering role of volunteering and its gendered patterns. Policymakers should offer more volunteering opportunities and training to these economically inactive groups to reduce their risk of mental issues.

## 1. Introduction

Research from multiple disciplines has confirmed that paid workers have better health than unemployed or economically inactive groups, regardless of age, gender, or social class [1,2,3]. This is because paid work provides individuals with psychosocial benefits (e.g., social contact, collective purpose, and status), thereby benefiting identity development, mental health, and well-being [2,4]. In contrast, unemployed and economically inactive groups are deprived of these psychosocial benefits, thereby having a higher risk of mental issues and worse well-being [3,5,6]. Given the negative impact of unemployment on mental health and well-being [7,8], volunteering has received substantial scholarly attention as a positive factor in promoting mental health. However, current findings are inconclusive as to whether volunteering can mitigate the negative impacts of unemployment and economic inactivity.

On the one hand, research suggests that volunteering can help mitigate the negative mental effects brought on by unemployment among the elderly by alleviating their anxiety and depression [9]. In addition, longitudinal evidence from Sweden reveals that volunteering during unemployment can significantly reduce the likelihood of smoking and drinking [10]. On the other hand, in countries with lower unemployment benefits, those who regularly participate in volunteering have worse mental health than non-volunteers [6]. Several limitations lead to these inconsistent findings in the previous studies. First, the majority of previous studies remain narrow in scope, focusing exclusively on the older age sample [9,11,12,13] while ignoring the working-age population. Moreover, there is a lack of discussion regarding the potential buffering benefits of volunteering for protecting the mental health of the economically inactive group and its subgroups. The health inequalities need to urgently be addressed, as the disadvantaged positions of the economically inactive groups lead to higher risks of mental issues. Accordingly, it is worthwhile to investigate the impact of volunteering on the mental health of a wider range of population groups that lack access to paid work.

### 1.1. Theoretical Bases

This section, based on Jahoda’s Latent Deprivation Theory [14] and Fryer’s Agency Restriction argument [14], aims to demonstrate the extent to which volunteering can mitigate the negative impacts of losing paid work on mental health.

First, Jahoda’s Latent Deprivation Theory supports the notion that volunteering can benefit mental health by offering psychosocial benefits. Specifically, Jahoda (1982) argues that paid employment, acting as a dominant social institution in contemporary society, is more than a source of income (e.g., manifest function) [13]. In other words, it provides a number of latent functions (e.g., time structure, enforced activity, social contact, collective purpose, social status, and identity) to fulfill individuals’ psychosocial needs. As an unpaid but productive activity in the public sphere [15], volunteering provides unemployed people with the opportunity to meet their psychosocial needs through alternative employment. Therefore, volunteering compensates in a non-fiscal way for the loss of potential paid-work benefits. Second, Fryer’s Agency Restriction argument indicates that although volunteering cannot fully emulate employment by providing essential financial resources, it can benefit mental health by promoting life control and self-efficacy. Specifically, Fryer (1986) argues that loss of income and lack of sense of control over one’s life can both deteriorate mental health and well-being during unemployment periods. Studies on time use suggest that inactive economic status can lead to a lack of temporal predictability, thereby creating a higher risk of unhealthy daily practices (e.g., less engagement in sports, irregular eating behaviors, irregular sleep, etc.) [16,17,18]. We assume that volunteering can be a buffer against the adverse impacts of inactive economic status since volunteering encourages people to schedule their time and actively engage in social contact [19,20]. Therefore, volunteering might be efficient in enhancing temporal predictability [18] of inactive economic groups and producing similar effects to those of the latent function of work.

Accordingly, although it is uncertain whether volunteering can completely offset the financial effects of unemployment, we expect that volunteering can counteract the negative mental effects of unemployment by offering latent benefits and promoting a sense of life control.

### 1.2. Current Study

This paper provides important insights into the benefits of volunteering on mental health across different employment status groups. Given the wide range of volunteering benefits for mental health and currently high levels of unemployment, this paper has important implications for how volunteering can address adverse effects on people’s health caused by labor-market instability. Furthermore, by analyzing gender differences, this paper provides a nuanced understanding of the impact of volunteering on the mental health of economically inactive groups. Overall, this article’s findings suggest that volunteering as alternative work can significantly buffer adverse mental issues during an economically inactive period. In addition, this article highlights the need for public health policymakers to adopt gender-specific approaches to promote opportunities and training of volunteering for economically inactive groups. For applications in public health, this study’s findings empirically suggest expanding applications of volunteering in addressing mental issues among elder groups to the whole of economically inactive groups.

## 2. Method

### 2.1. Data and Sample

This study uses longitudinal panel data on employment and health outcomes from five waves (2010–2020) of the UK Household Longitudinal Study (UKHLS), which consists of a stratified and clustered general population sample of roughly 40,000 households. This article uses the second, fourth, sixth, eighth, and tenth waves of the UKHLS, as they include information on volunteering behavior and frequency of respondents who are employed, unemployed, and economically inactive (e.g., home care, long-term illness/disability, and unpaid work). The final analytical sample consisted of 127,405 individual wave observations, with 79.71% employed, 6.48% unemployed, and 13.81% of economically inactive (e.g., home care, long-term illness/disability, and unpaid work). The sample sizes for the second, fourth, sixth, eighth, and tenth waves were 30,529, 27,213, 24,164, 24,353, and 21,146, respectively. The UKHLS longitudinal weights were used to adjust for the complex survey design, non-response rate, unequal selection probabilities, and attrition over waves.

### 2.2. Measurements

#### 2.2.1. Dependent Variable

*Mental health* is measured by the 12-item General Health Questionnaire (GHQ-12), a widely used and reliable measure of mental health [21]. The answers to the GHQ-12 were converted to a single continuous scale ranging from 0 (the least distressed) to 36 (the most distressed) in the UKHLS. Consistent with previous studies, this research reverses the scale for the convenience of results interpretation, in which the higher the score, the better the mental health [3].

#### 2.2.2. Independent and Moderator Variables

*Current economic activity* is the key independent variable consisting of the following categories: employed, unemployed, family caregivers, long-term sick, and doing unpaid work.

*Volunteering behavior* is the main moderator measured by respondents’ answers to whether they have engaged in any unpaid volunteering activity or volunteered for any organization or charity within the last 12 months.

*Volunteering frequency* is measured by respondents’ answers to how often they engaged in volunteering over the last 12 months, scaling from 1 (on three or more days a week) to 9 (on a seasonal basis). We dichotomized their answers to: (less frequent) once per month or more, and (frequent) less than once per month.

#### 2.2.3. Confounders

All the analyses in the research controlled a series of time-varying confounders identified by the previous studies [5,9], including age (grand mean-centered), age squared, marital status, presence of children, long-term illness, and logarithmic household income. Wave dummies were also controlled to capture any individual-level idiosyncratic disturbances over time. All variables used in this study are described in Table 1.

### 2.3. Analytic Approach

This study uses a fixed effects (FE) regression model, which has two advantages over cross-sectional analysis. First, FE regression on within-individual variation not only eliminates unobserved heterogeneity—a confounding effect that occurs in all time-invariant variables—but also reduces the bias that is derived from between-individual comparisons [22]. Second, the FE model uses lagged independent variables in the panel data, which are beneficial with respect to understanding the dynamic influence of economic activity and voluntary participation on mental health [22]. Given the wide range of differences in economic activity, voluntary participation, and frequency between men and women [23,24], separate FE models are fitted by gender and controlled for other sociodemographic characteristics.

## 3. Results

Table 2 shows several FE models that predict the effects of current economic activity statuses, volunteering behavior, and volunteering frequency on mental health. First, as shown in Model 1, volunteering behavior is significantly associated with better mental health (β = 0.17, *p* < 0.001), while unemployment and other types of economic inactivity are significantly associated with worse mental health. Second, Model 2 shows that the frequency of volunteering matters. Specifically, participating in volunteering once per month or more is significantly associated with better mental health (β = 0.22, *p* < 0.001), while less than once per month is insignificant. Third, further analyses examine the interactions between each current economic activity and volunteering behavior. Model 3 indicates that volunteering behavior only significantly interacted with family care. People who participate in volunteering have better mental health than non-volunteers during family care (β = 0.51, *p* < 0.01). Fourth, after considering the volunteering frequency, Model 4 shows that participating in volunteering once per month or more is anticipated to have better mental health for those during family care (β = 0.65, *p* < 0.001) and long-term sick (β = 1.17, *p* < 0.01).

Taken together, the results in Table 2 generally suggest that all economic inactivity have adverse impacts on mental health. Consistent with our expectations, volunteering can buffer the adverse mental effects of some economic inactivity, including family care and long-term sickness. However, volunteering cannot mitigate the negative mental effects of unemployment and unpaid work. It is worth noting that the frequency of volunteering matters. Specifically, unless participating in volunteering once per month or more, volunteering cannot significantly mitigate the adverse mental effects of any unemployment and economic inactivity.

Further analyses explore the potential gender patterns of direct and buffering impacts of volunteering.Table 3 reports a series of FE models that predict the effects of economic activity and voluntary participation on the mental health of men and women. First, Models 5 and 7 indicate that volunteering behavior (regardless of its frequency) has no significant impact on the mental health of men. In contrast, as shown in Models 6 and 8, both volunteering behavior and frequent volunteering have positive impacts on the mental health of women. Therefore, the direct mental benefits of volunteering are only significant in the women sample. The adverse effects of each type of economic inactivity remain significant after separately analyzing by gender sample. Second, Models 9 and 11 test the interactions between each economic activity and volunteering behavior or frequency for men. Specifically, as Model 9 shows, volunteering behavior can significantly buffer the adverse effects of long-term sickness for men. Additionally, Model 11 shows that frequent participation in volunteering can significantly buffer the adverse effects of family care and long-term sickness for men. Third, Models 10 and 12 test the interactions between each economic activity and volunteering for women. Specifically, engaging in volunteering (especially frequent volunteering) can significantly reduce the adverse effects of family care and unpaid work for women.

Further, we plot Figure 1 (based on Model 11) and Figure 2 (based on Model 12) to illustrate the gender patterns of volunteering’s buffering effects. Frequent voluntary work can have a significant positive effect on the mental health of men who are economically inactive due to family care or long-term illness. Nevertheless, there is no compelling evidence that volunteering buffers the adverse effects of the other two types of economic inactivity. As illustrated in Figure 2, only women who are economically inactive due to family care commitments or other unpaid work experience any mental health benefits from frequent voluntary work.

Overall, we found that volunteering can help mitigate the negative impacts of some types of economic inactivity, but benefits vary by gender. For men, frequent volunteering can buffer the adverse effects of family care and long-term illness. For women, frequent volunteering can buffer the adverse effects of family care and unpaid work. Inconsistent with our expectations, volunteering (regardless of frequency) cannot buffer the adverse mental effects brought by unemployment. These findings suggest that the positive mental health effects of volunteering are different across economic activity types and genders.

## 4. Discussion

Given the wide range of volunteering benefits for mental health and trends in economic inactivity, it is important to analyze how the act of volunteering can address the negative health effects brought on by labor-market instability. By analyzing the relationships between volunteering, economic activity, and mental health, we find that the low mental health status of various economic inactive groups can be buffered by volunteering, and that the pattern varies across genders. Overall, there are three key findings in this study.

First, inconsistent with our expectations, volunteering cannot mitigate the negative impact of unemployment on mental health. Although we would expect this to be true based on existing research [25,26], the empirical results do not support this hypothesis. This finding indicates the positive effect of paid work on mental health and the fact that volunteering does not fully address the psychological problems caused by unemployment.

Second, volunteering can buffer the adverse effects of some types of economic inactivity. This finding supports our expectations that volunteering might serve as social support for people who cannot participate in the workforce. First, for those with long-term sickness, volunteering can compensate for their physical health disadvantages and enhance their self-efficacy. Second, for family caregivers or unpaid workers, volunteering provides an opportunity to promote their social engagement, thereby increasing their sense of life control and their mental health [27]. It is worth noting that the frequency of volunteering matters. Specifically, there are no significant differences in terms of mental health between those who have never volunteered and those who volunteer less than once a month. Therefore, people cannot benefit from the buffering effects of volunteering unless they participate in volunteering at least once per month. It is also worth noting that most of the current findings on the buffering role of volunteering are focused on the elder group [9,11,12], while they might not be able to keep a stable high frequency of engaging in volunteering due to limited energy and physical strength. Hence, it is necessary to explore how to promote some particular pathways of volunteering for the elderly.

Finally, the impact of volunteering on the mental health of economically inactive groups differs significantly by gender. Volunteering appears to be perceived as a temporary and transitional process for men until they find formal paid employment, and therefore does not provide mental health benefits equivalent to those gained from employment. However, for women, regardless of whether they are family caregivers or doing unpaid work, volunteering can significantly improve their mental health. Previous studies have found gender disparities in the mental benefits of volunteering [28,29,30], while our findings further examine the gender patterns of the buffering role of volunteering. These findings contribute insights into the identification of the potential social barriers to promoting and operating volunteering activities [29].

This study has a number of limitations. First, although we used longitudinal data and FE models, caution should be exercised in interpreting the results as causal due to the possible omission of time-varying variables, e.g., changes in social security resources. Second, though volunteering participation improves the mental health of populations such as family caregivers, the explanatory mechanisms are currently unknown. Future studies with qualitative methods such as ethnographies and interviews are required. Third, UK data were used in this study, and, as such, the mental health of unemployed people who volunteer may be dependent on the national welfare system [3,6]. Thus, this needs to be further tested by using data across countries. Fourth, due to the measurement bias of self-reported mental health indicators, we suggest that future studies examine some objective mental indicators (e.g., allostatic load) [31,32] to explore the benefits of volunteering better.

## 5. Conclusions

Volunteering was found to buffer the negative mental effects of inactive economic status. In addition, the buffering effects of volunteering depend on the frequency of engagement and vary across gender. Engaging in volunteering activities once per month or more can buffer the negative impacts of family care or long-term sickness for men and the negative impacts of unpaid work for women. Therefore, it is necessary to promote volunteering engagement as a social interference for protecting the mental health of the inactive economic groups.

## Figures and Tables

**Figure 1 ijerph-19-06809-f001:**
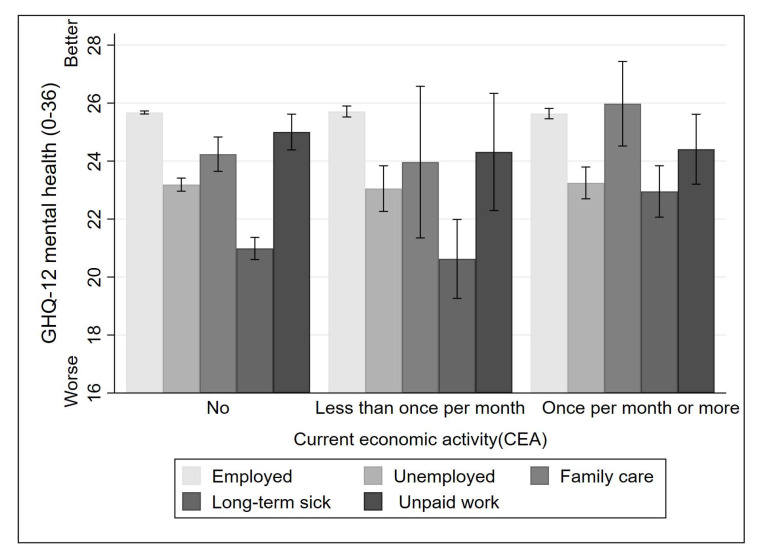
Differences in mental health between volunteering frequency distinguish current economic activity (Men).

**Figure 2 ijerph-19-06809-f002:**
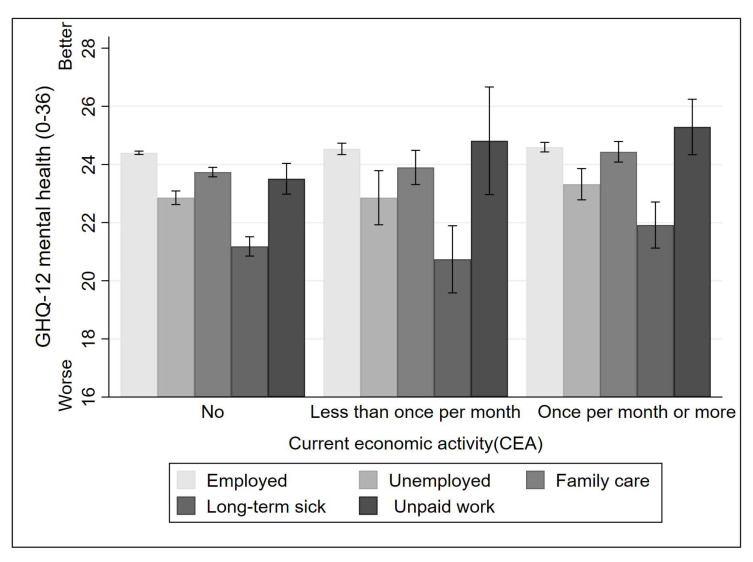
Differences in mental health between volunteering frequency distinguish current economic activity (Women).

**Table 1 ijerph-19-06809-t001:** Unweighted descriptive statistics by gender (UKHLS 2010–2020).

	Men	Women	T/Z Tests *p*-Values
GHQ-12 (M, SD)	25.31 (5.30)	24.15 (5.95)	<0.001
Current economic activity(CEA)(%)			<0.001
Employed	85.75	74.88	
Unemployed	7.68	5.52	
Family care	0.93	13.70	
Long-term sick	4.64	4.83	
Unpaid work	1.01	1.07	
Volunteer behavior(VB)(%)			<0.001
No	83.53	81.92	
Yes	16.47	18.08	
Volunteer frequency(VF)(%)			<0.001
No	83.53	81.92	
Less than once per month	6.15	5.84	
Once per month or more	10.32	12.24	
Age (M, SD)	43.26 (13.19)	42.52 (12.67)	<0.001
Marital status(%)			<0.001
Never married	22.21	20.24	
Married	71.49	67.21	
Divorced/separated/widowed	6.30	12.55	
Have Children(%)			<0.001
No	61.77	55.64	
Yes	38.23	44.36	
Have Long-standing illness(%)			<0.001
No	72.15	69.65	
Yes	27.85	30.35	
Subjective financial situation(%)			<0.001
Living comfortably	27.25	25.38	
Doing alright	38.02	37.93	
Just about getting by	24.55	25.43	
Finding it quite difficult	7.03	7.8	
Finding it very difficult	3.06	3.35	
Missing value	0.09	0.1	
Logged household income (M, SD)	7.55 (0.79)	7.49 (0.73)	<0.001
N (person-wave observations)	56,611	70,794	

Note. Proportions reported for categorical variables. Mean values reported for continuous variables. Standard deviations are in parentheses.

**Table 2 ijerph-19-06809-t002:** Two-way fixed effects models predicting the effects of current economic activity type and voluntary participation on mental health (reversed GHQ-12).

	Model 1	Model 2	Model 3	Model 4
Current economic activity (ref. = Employed)				
Unemployed	−1.90 *** (0.11)	−1.90 *** (0.11)	−1.93 *** (0.12)	−1.93 *** (0.12)
Family care	−0.72 *** (0.09)	−0.72 *** (0.09)	−0.81 *** (0.10)	−0.81 *** (0.10)
Long-term sick	−3.66 *** (0.20)	−3.66 *** (0.20)	−3.76 *** (0.21)	−3.77 *** (0.21)
Unpaid work	−0.56 ** (0.20)	−0.56 ** (0.20)	−0.74 ** (0.23)	−0.74 ** (0.23)
Volunteer behavior	0.17 *** (0.05)		0.09 + (0.05)	
Volunteer frequency (ref. = No)				
Less than once per month		0.10 (0.07)		0.09 (0.07)
Once per month or more		0.22 *** (0.06)		0.10 (0.06)
Current economic activity × Volunteer behavior				
Unemployed × Yes			0.16 (0.23)	
Family care × Yes			0.51 ** (0.17)	
Long-term sick × Yes			0.70 + (0.36)	
Unpaid work × Yes			0.67 + (0.41)	
Interaction: Current economic activity × Volunteer frequency				
Unemployed × Less than once per month				−0.04 (0.44)
Unemployed × Once per month or more				0.24 (0.25)
Family care × Less than once per month				0.06 (0.32)
Family care × Once per month or more				0.65 *** (0.19)
Long-term sick × Less than once per month				−0.43 (0.63)
Long-term sick × Once per month or more				1.17 ** (0.41)
Unpaid work × Less than once per month				0.31 (0.76)
Unpaid work × Once per month or more				0.79 + (0.44)
Constant	22.06 *** (0.59)	22.05 *** (0.59)	22.08 *** (0.59)	22.07 *** (0.59)
Waves dummies	Yes	Yes	Yes	Yes
Observations	127,405	127,405	127,405	127,405
Number of respondents	47,607	47,607	47,607	47,607
*R* ^2^	0.027	0.027	0.027	0.027

Note. All models controlled for age, age squared, marital status, presence of children, household income, and presence of longstanding illness. Robust standard errors in parentheses. + *p* < 0.1, ** *p* < 0.01, *** *p* < 0.001.

**Table 3 ijerph-19-06809-t003:** Two-way fixed effects models predicting the effects of current economic activity type and voluntary participation on mental health (reversed GHQ−12) for men and women.

	Model 5	Model 6	Model 7	Model 8	Model 9	Model 10	Model 11	Model 12
	Men	Women	Men	Women	Men	Women	Men	Women
Current economic activity (ref. = employed)								
Unemployed	−2.46 *** (0.16)	−1.48 *** (0.15)	−2.46 *** (0.16)	−1.48 *** (0.15)	−2.47 *** (0.16)	−1.53 *** (0.16)	−2.47 *** (0.16)	−1.53 *** (0.16)
Family care	−1.24 *** (0.35)	−0.59 *** (0.10)	−1.25 *** (0.35)	−0.59 *** (0.10)	−1.43 *** (0.38)	−0.66 *** (0.11)	−1.44 *** (0.38)	−0.66 *** (0.11)
Long-term sick	−4.53 *** (0.31)	−3.15 *** (0.26)	−4.53 *** (0.31)	−3.15 *** (0.26)	−4.70 *** (0.33)	−3.19 *** (0.28)	−4.70 *** (0.33)	−3.20 *** (0.28)
Unpaid work	−0.74 * (0.31)	−0.46 + (0.26)	−0.74 * (0.31)	−0.46 + (0.26)	−0.55 (0.35)	−0.86 ** (0.31)	−0.54 (0.35)	−0.87 ** (0.31)
Volunteer behavior	0.04 (0.07)	0.26 *** (0.07)			−0.01 (0.08)	0.17 * (0.07)		
Volunteer frequency(ref. = no)								
Less than once per month			0.02 (0.10)	0.15 (0.10)			0.03 (0.10)	0.14 (0.10)
Once per month or more			0.06 (0.09)	0.33 *** (0.08)			−0.04 (0.10)	0.20 * (0.09)
Current economic activity × Volunteer behavior								
Unemployed × Yes					0.07 (0.30)	0.20 (0.33)		
Family care × Yes					1.29 + (0.69)	0.39 * (0.18)		
Long-term sick × Yes					1.37 * (0.57)	0.23 (0.46)		
Unpaid work × Yes					−0.72 (0.70)	1.49 ** (0.50)		
Interaction: Current economic activity × Volunteer frequency								
Unemployed × Less than once per month							−0.05 (0.56)	0.02 (0.69)
Unemployed × Once per month or more							0.15 (0.35)	0.25 (0.36)
Family care × Less than once per month							−0.28 (1.23)	0.01 (0.33)
Family care × Once per month or more							1.79 * (0.75)	0.50 * (0.20)
Long-term sick × Less than once per month							−0.37 (1.05)	−0.49 (0.79)
Long-term sick × Once per month or more							2.02 ** (0.62)	0.56 (0.54)
Unpaid work × Less than once per month							−0.94 (1.40)	1.22 (0.83)
Unpaid work × Once per month or more							−0.62 (0.74)	1.56 ** (0.55)
Constant	23.88 *** (0.74)	20.69 *** (0.86)	23.88 *** (0.74)	20.68 *** (0.86)	23.91 *** (0.74)	20.70 *** (0.86)	23.90 *** (0.74)	20.69 *** (0.86)
Waves dummies	Yes	Yes	Yes	Yes	Yes	Yes	Yes	Yes
Observations	56,611	70,794	56,611	70,794	56,611	70,794	56,611	70,794
Number of respondents	21,721	25,886	21,721	25,886	21,721	25,886	21,721	25,886
*R* ^2^	0.035	0.023	0.035	0.023	0.035	0.024	0.036	0.024

Note. All models controlled for age, age squared, marital status, presence of children, household income, and presence of longstanding illness. Robust standard errors in parentheses. + *p* < 0.1, * *p* < 0.05, ** *p* < 0.01, *** *p* < 0.001.

## Data Availability

Data available from an open-access public depository (accessible at https://www.understandingsociety.ac.uk), accessed on 1 May 2022.

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
