# Peer review of "Can Volunteering Buffer the Negative Impacts of Unemployment and Economic Inactivity on Mental Health? Longitudinal Evidence from the United Kingdom"

_ijerph, 2022, doi:10.3390/ijerph19116809_

Round 1
Reviewer 1 Report
Dear authors,
Thank you for allowing me to read the manuscript addressing whether volunteering can buffer the negative impacts of unemployment and economic inactivity on mental health. This is particularly the case with the United Kingdom, which has been analyzed in your study.
As far as I understood from the paper, your research aims to provide an interesting approach to how volunteering can address the adverse effects on people’s health caused by labor-market instability through a fixed effects (FE) regression model based on the data gathered from the United Kingdom Longitudinal Household Study (2010-2020).
It appears that the research has been well designed and the paper is properly drafted in English with good academic soundness. However, some aspects of your research approach should be improved a little bit to better understand the whole manuscript. I hope your efforts in the revision of the manuscript according to my comments can make the paper more attractive to satisfy the high demands set by readers of the International Journal of Environmental Research and Public Health (ISSN 1660-4601),
#Abstract:
No objections.
#Introduction:
This part may be very short and does not help readers to understand the whole framework of the research approach, since no matters directly concerning health economics have been considered in the approach. I urge you to expand it a bit for a broader explanation of the approach of your study.
#Method:
This paragraph is a key part of the manuscript. I also think that Table 1 is very valuable since they help to gain a deeper understanding of particular issues from your research.
#Results:
This subpart is good. Particularly commendable is the creation of Tables 2 and 3. That´s why I do not have anything negative to say about it. Moreover, Figures 1 and 2 are very helpful for readers, since they have summarized the most important numerical findings of the research.
#Discussion and conclusions:
This section seems too short for the ambitious objectives pointed out in the introduction. Please try to expand it to better understand the findings from this study and how they may provide a basis for further work. I truly believe that you should devote some paragraphs to consideration of matters relating to conclusions in a specific subchapter that would be separated from the discussion.
Additionally, I suggest that a separate section should be devoted to explaining your research recommendations for future studies.
Finally, I really cannot see any relationship between the findings of your study and the aims thereof in terms of public health. In fact, in all subchapters of the manuscript, the term “public health” is not once mentioned. Therefore, I encourage you to review the content of your findings to favor an approach toward public health.
Best Regards,
The Reviewer
Author Response
Dear reviewer,
Re: Manuscript ID: ijerph-1731732 and Title: “Can volunteering buffer the negative impacts of unemployment and economic inactivity on mental health? Longitudinal evidence from the United Kingdom.”
Thank you for your comments concerning our manuscript. Those comments are valuable and very helpful. We have read through the comments carefully and have made corrections. As the editor suggested, revisions in the text are highlighted by using track changes. The details of our response are as follows:
Comment 1
“#Introduction:
“This part may be very short and does not help readers to understand the whole framework of the research approach, since no matters directly concerning health economics have been considered in the approach. I urge you to expand it a bit for a broader explanation of the approach of your study”.
Response: We appreciate the reviewer’s suggestion. We expanded the introduction by adding more information about the assumptions’ theoretical bases (page 3) and highlighting the connections between our study’s findings and public health (pages 2 & 4).
Comment 2
“#Discussion and conclusions:
This section seems too short for the ambitious objectives pointed out in the introduction. Please try to expand it to better understand the findings from this study and how they may provide a basis for further work. I truly believe that you should devote some paragraphs to consideration of matters relating to conclusions in a specific subchapter that would be separated from the discussion.”.
Response: We appreciate the reviewer’s suggestion. The discussion of the study’s future implications is too short to be understood. To address the reviewer’s concern, we now have expanded the discussion (page 9) and added a conclusion (page 10) to emphasise the future implications of the study’s findings in addressing the health inequalities in public health.
Comment 3
“I suggest that a separate section should be devoted to explaining your research recommendations for future studies.”.
Response: We appreciate the reviewer’s suggestion. We have separated the section into discussion and conclusion and expanded the recommendations for future studies. For example, we emphasise the recommendations for future studies in exploring the associations between volunteering and mental health by using objective mental indicators, such as allostatic loads (page 10).
Comment 4
“I really cannot see any relationship between the findings of your study and the aims thereof in terms of public health. In fact, in all subchapters of the manuscript, the term “public health” is not once mentioned. Therefore, I encourage you to review the content of your findings to favor an approach toward public health.”
Response: We appreciate the reviewer’s suggestion. We have thoroughly added more discussion of the study’s objectives and contributions to emphasise how the study’s findings contribute to understanding the potential health inequalities in public health (pages 2 & 4 & 9 & 10). In addition, we highlighted the study’s contribution to public health in the final section (conclusion) (page 10).
Reviewer 2 Report
The reviewed article presents an important issue of the impact of volunteering on the mental health. The discussed topic requires further scientific exploration.
The purpose of the article should be clearly defined. It requires improvements.
I suggest supplementing the subject literature and expanding the introduction and literature review based on the latest publications from this area.
I recommend discussing the research results based on the latest research achievements in this field.
I suggest extending the conclusions. Concluding remarks should be clear and comprehensive.
I recommend checking certain aspects in the bibliographic reference lists. For example, the name of the journal must be written with its abbreviation, the volume must be written in italics and the year in bold. It requires improvements.
Author Response
Dear reviewer,
Re: Manuscript ID: ijerph-1731732 and Title: “Can volunteering buffer the negative impacts of unemployment and economic inactivity on mental health? Longitudinal evidence from the United Kingdom.”
We appreciate your comments and suggestions concerning our manuscript. Those comments are constructive. We have read through the comments carefully and have made corrections. As the editor suggested, revisions in the text are highlighted by using track changes. The details of our response are as follows:
Comment 1
“The purpose of the article should be clearly defined. It requires improvements”.
Response: Many thanks for the reviewer’s suggestions. To address the reviewer’s concern, we have added further information in the discussion (page 9) and abstract (page 1) about our study's implications and theoretical contributions to highlight the purpose of our study.
Comment 2
“I suggest supplementing the subject literature and expanding the introduction and literature review based on the latest publications from this area”.
Response: We appreciate the reviewer’s suggestion. We have added further theoretical arguments and empirical findings to provide a more detailed description of the assumption about the potential buffering role of volunteering. Following Fryer’s arguments about the importance of the sense of life control, we discussed the evidence from studies on time use (page 3 bottom), which further explains how volunteering might improve mental health by ensuring a healthy temporal schedule.
Comment 3
“I recommend discussing the research results based on the latest research achievements in this field”.
Response: We appreciate the reviewer’s suggestion that providing more information on the latest relevant research findings can improve the discussion part of our article. In the discussion section, we have now discussed more current findings on the potential social barriers to volunteering and the gender disparities in the mental benefits of volunteering (page 9). The further discussion emphasises some new contexts for studying volunteering, especially for potentially disadvantaged groups.
Comment 4
“I suggest extending the conclusions. Concluding remarks should be clear and comprehensive”
Response: We appreciate the reviewer’s suggestions. We have added the conclusion part to emphasise our findings and contributions (page 9). In addition, we emphasise the future implications of our findings in the abstract (page 1)
Comment 5
“I recommend checking certain aspects in the bibliographic reference lists. For example, the name of the journal must be written with its abbreviation, the volume must be written in italics and the year in bold. It requires improvements”
Response: We appreciate the reviewer’s suggestions. We have now formatted the references and citations to follow the journal’s guidelines.
Reviewer 3 Report
Dear authors,
The topic of analysis is interesting but the manuscript needs improvement.
First of all, I would suggest that you reread the editing conditions imposed by the journal and include them in your text.
From my point of view, I would need more information about selecting the data and explaining the variables.
The conclusions and future implications of your study can be improved and better highlighted.
Best regards.
Author Response
Dear reviewer,
Re: Manuscript ID: ijerph-1731732 and Title: “Can volunteering buffer the negative impacts of unemployment and economic inactivity on mental health? Longitudinal evidence from the United Kingdom.”
We appreciate your comments and suggestions concerning our manuscript. Those comments are valuable and constructive. We have read through the comments carefully and have made corrections. As the editor suggested, revisions in the text are highlighted by using track changes. The details of our response are as follows:
Comment 1
“First of all, I would suggest that you reread the editing conditions imposed by the journal and include them in your text”.
Response: We appreciate the reviewer’s suggestions. We have thoroughly checked the journal’s guidelines and then modified and formatted the article to meet the requirements.
Comment 2
“From my point of view, I would need more information about selecting the data and explaining the variables”.
Response: We appreciate the reviewer’s suggestions. We have modified and added more information about selecting the data (page 4) and expanded more financial situations in Table 1 (page 13).
Comment 3
"The conclusions and future implications of your study can be improved and better highlighted."
Response: We appreciate the reviewer’s suggestions. We have now added a new paragraph for concluding our findings and future implications (page 9). We also highlighted the future implications in the abstract.
Reviewer 4 Report
Dear Author(s),
Thank you for your paper. I would like to underline that I found the paper interesting and with few aspects and elements of novelty.
These are revisions suggested, as I believe this paper would be a great fit for the journal:
Lines 35- 37: suggest “ This is because paid work provides 35 individuals with psychosocial benefits (e.g., social contact, collective purpose, and status), 36 thereby benefiting their identity development, mental health, and wellbeing (Fryer, 1986; 37 Jahoda, 1982; Wood & Burchell, 2018). “ by using Before-pandemic and remote working studies. Which makes conclusions outdated with the risk mislead and a very low level of novelty.
71 loss of potential paid-work benefits.
98 Impressive sample size. “general population sample of roughly 40,000 households”. However, is not clear about the socio-economic level of respondents. It is highly recommended to take a deeper look at respondent’s economic comfort since the study mentions “loss of potential paid-work benefits” (line 71)
102 states “25,000 employed, unemployed and economically inactive”. could be relevant to mention the % of employed respondents. And table1 mentions that more than 75% of the respondents are employed. To simplify reading and avoid confusing interpretation, authors should consider only the population considered for the study purpose and give a clear idea about the sample size.
In general, it is highly suggested to make more careful conclusions and use the sample. Otherwise, for example, Figure 1, suggests that “Once per month family care” is healthier than being employed. What use society should make of this type of conclusion?
Thank you again for your work and I wish you all the best in your future research!
Author Response
Dear reviewer,
Re: Manuscript ID: ijerph-1731732 and Title: “Can volunteering buffer the negative impacts of unemployment and economic inactivity on mental health? Longitudinal evidence from the United Kingdom.”
Many thanks for your comments and suggestions concerning our manuscript. Those comments are valuable and very helpful. We have read through the comments carefully and have made corrections. As the editor suggested, revisions in the text are highlighted by using track changes. The details of our response are as follows:
Comment 1
Lines 35- 37: suggest “ This is because paid work provides 35 individuals with psychosocial benefits (e.g., social contact, collective purpose, and status), 36 thereby benefiting their identity development, mental health, and wellbeing (Fryer, 1986; 37 Jahoda, 1982; Wood & Burchell, 2018). “ by using Before-pandemic and remote working studies. Which makes conclusions outdated with the risk mislead and a very low level of novelty.
Response: We appreciate the reviewer’s suggestions for improving the novelty of the cited works. To address the reviewer’s concern, we have recited some recent works (from 2019 to 2022) on the benefits of paid work (page 2 highlights).
Comment 2
“general population sample of roughly 40,000 households”. However, is not clear about the socio-economic level of respondents. It is highly recommended to take a deeper look at respondent’s economic comfort since the study mentions “loss of potential paid-work benefits” (line 71).
Response: We appreciate the reviewer’s suggestions. We have now added subjective financial situation information to the descriptive statistics of the respondents (Table 1).
Comment 3
“102 states “25,000 employed, unemployed and economically inactive”. could be relevant to mention the % of employed respondents. And table1 mentions that more than 75% of the respondents are employed. To simplify reading and avoid confusing interpretation, authors should consider only the population considered for the study purpose and give a clear idea about the sample size..
Response: We appreciate the reviewer’s suggestions for simplifying reading and avoiding confusing interpretations. We have now added a description of the employment proportion and sample size of the final analytical sample (page 4).
Comment 4
“In general, it is highly suggested to make more careful conclusions and use the sample. Otherwise, for example, Figure 1, suggests that “Once per month family care” is healthier than being employed. What use society should make of this type of conclusion?."
Response: We appreciate the reviewer’s suggestions that the confusing interpretations of the information from the figures make it difficult for readers to understand our conclusions. We have now improved the interpretations of the results (especially for the information in our figures) (page 7).
Round 2
Reviewer 3 Report
Dear authors,
Thank you for your efforts to include my comments and suggestions in the manuscript.
I wish you much success in the future!
Best regards,